

# Polymethoxylated flavonoids in citrus fruits: absorption, metabolism, and anticancer mechanisms against breast cancer

Yiyu Wang[1], Yuan Mou[2], Senlin Lu[1,3], Yuhua Xia[1] and Bo Cheng[4]

[1] Hubei Key Laboratory of Tumor Microenvironment and Immunotherapy, China Three Gorges University, Yichang, Hubei, China
[2] Department of General Surgery, People's Hospital Affiliated to Chongqing Three Gorges Medical College, Wanzhou District, Chongqing, China
[3] Chong Qing Wan Zhou Health Center for Women and Children, Wanzhou, Chongqing, China
[4] Xinjiang Institute of Materia Medica, Key Lab of Xinjiang Uighur Medicine, Urumqi, Xinjiang, China

## ABSTRACT

Polymethoxylated flavonoids (PMFs) are a subclass of flavonoids found in citrus fruits that have shown multifunctional biological activities and potential anticancer effects against breast cancer. We studied the absorption, metabolism, species source, toxicity, anti-cancer mechanisms, and molecular targets of PMFs to better utilize their anticancer activity against breast cancer. We discuss the absorption and metabolism of PMFs in the body, including the methylation, demethylation, and hydroxylation processes. The anticancer mechanisms of PMFs against breast cancer were also reviewed, including the estrogen activity, cytochrome P-450 enzyme system, and arylhydrocarbon receptor (AhR) inhibition, along with various molecular targets and potential anticancer effects. Although PMFs may be advantageous in the prevention and treatment for breast cancer, there is a lack of clinical evidence and data to support their efficacy. Despite their promise, there is still a long way to go before PMFs can be applied clinically.

# INTRODUCTION

Polymethoxylated flavonoids (PMFs) are commonly found in citrus fruits belonging to the Rutaceae family. They are secondary metabolites of plants and have a basic structure derived from the flavone nucleus (*Ortuño et al., 2002*). PMFs have a low polarity and are created when two or more hydroxyl groups on the A and B rings of the flavone nucleus are replaced with two or more methoxy groups. Depending on the location and number of methoxy substitutions, PMFs can be categorized into different types. In the 1990s, PMFs were discovered to have anti-inflammatory and antiallergic effects when extracted from immature citrus peels (*Kim et al., 1999*). Over the years, scientists have become increasingly interested in dietary flavonoids, including PMFs due to their physiological activities, including their anti-inflammatory and immune effects, as well as their metabolic,

Corresponding authors
Yiyu Wang, wang_yiyu11@163.com
Bo Cheng, chengbo0417@163.com

**Figure 1** Structure of PMFs with anti-breast cancer activity.

cardiovascular, neurodegenerative, and cancer disease prevention properties (*Cassidy & Minihane, 2017*; *Ahmed et al., 2021*).

Breast cancer has become the most prevalent cancer among women worldwide, with an incidence rate of 11.7%, according to a 2020 global epidemiological survey (*Sung et al., 2021*). As early as 1966, *So et al. (1996)* discovered that orange juice could inhibit the proliferation of human breast cancer cells and delay the occurrence of breast tumors (*So et al., 1996*). Recent studies have found that PMFs, such as nobiletin and tangeretin, exhibit dose- and time-dependent anticancer activities against different subtypes of breast cancer cell lines (*Morley, Ferguson & Koropatnick, 2007a*; *Chen et al., 2014b*). Similarly, *Hu et al. (2022)* found that precipitates extracted from citrus vinegar fermentation had effective anti-proliferation effects on MCF7 cells by inhibiting microtubule protein polymerization (*Hu et al., 2022*). Most unmethylated dietary flavonoids have low bioavailability, but methylated dietary flavonoids not only retain the anticancer activity of unmethylated dietary flavonoids, but also have higher water solubility, metabolic stability, bioavailability, and lower toxicity (*Walle et al., 2007*). The majority of PMFs are found in citrus plants, with nobiletin, tangeretin, and sinensetin being the most abundant and widely studied PMFs in citrus peel (*Lai et al., 2015*). This study focused on 10 identified PMFs that have been reported to have anti-breast cancer effect (Fig. 1).

In recent years, there has been growing interest in the potential benefits of drugs that can be used for both medicinal and dietary purposes, such those that are widely available with low toxicity. Many researchers have focused on PMFs derived from dietary sources,
particularly those found in citrus plants, for their potential use in breast cancer treatment. However, a systematic review of these studies is currently lacking.

There are many reviews on the biological activities of compounds related to flavonoids, specifically their anticancer properties. However, there is a lack of anticancer activity of specific PMFs subclasses, especially for breast cancer, which is very common in women. This review systematically categorizes and discusses the anticancer effects and molecular mechanisms of various PMFs against breast cancer, including estrogen activity, arylhydrocarbon receptor inhibition, multidrug resistance protein inhibition, and chemical sensitization. The absorption and metabolism of 10 types of PMFs are introduced and the role of methylation in their metabolism is explained. The plant species sources, extraction methods, identification tools, and toxicity information are detailed. Lastly, the potential anticancer effects of PMFs on hormone-dependent cancers and the application of novel drug delivery systems were discussed. The aim of this article is to provide comprehensive insights into the anticancer effects of PMFs on breast cancer and to assist in better utilizing PMFs as chemopreventive and therapeutic agents for breast cancer.

## SURVEY METHODOLOGY

A variety of PMFs have been found to have certain anti-breast cancer effects in related experimental studies, including in $ER^+$ breast cancer and triple-negative breast cancer. We searched the commonly used literature databases. PubMed and Google Scholar, for the keywords "PMFs" and "breast cancer". We received few closely related responses so the literature database was expanded and the keywords were changed multiple times. However, such a simple search was not able to fully cover the relevant topics. Therefore, the PubChem database was used to classify PMFs by the number of methoxy groups and there were used as improved search keywords. After determining the subclass of PMFs, further determination of specific compound types in the subclass was made, and we then returned to the literature database for further retrieval.

### Anti-breast cancer molecular targets and mechanism of PMFs
#### P-450 enzyme inhibitor

The P-450 enzyme system has a significant role in activating many carcinogenic or pre-carcinogenic substances and in activating or deactivating anticancer drugs (*Venitt, 1994*). CYP enzymes can serve as genetic toxic intermediates, and their single nucleotide polymorphisms are linked to the risk of several cancers such as lung, liver, and head and neck cancers, but no evidence suggests a connection with breast cancer (*Agundez, 2004*). Later studies found that the single nucleotide polymorphisms of CYP enzymes were not associated with the risk of breast cancer (*Masson et al., 2005*; *Kiruthiga et al., 2011*; *Alanazi et al., 2015*; *Luo et al., 2021*). This indicates that the impact of the P-450 enzyme system on breast cancer occurrence and development may primarily focus on the metabolic activation or deactivation of xenobiotics (carcinogens and anticancer agents).

Polyaromatic hydrocarbons, including PMFs, can significantly reduce the activation of carcinogens and binding to DNA by affecting CYP1A1/1B1 transcription (*Walle, 2007a*). Nobiletin inhibits the induction of CYP1A1 and CYP1B1, increases its own metabolism,

and enhances proliferation inhibition in MCF7 and MDA-MB-468 breast cancer cells (*Surichan et al., 2012*). Tangeretin inhibits breast cancer cell proliferation through 4′-hydroxylation mediated by CYP1A1/CYP1B1 metabolites. In human breast cancer cells expressing CYP1 (MDA-MB-468), nobiletin, tangeretin, and sinensetin are metabolized into single demethylated derivatives, while normal human breast cells that do not express CYP1 (MCF10A) have no related metabolites. Nobiletin, tangeretin, and sinensetin inhibit proliferation to a certain degree in MDA-MB-468 cells but have no significant cytotoxicity on normal cells. CYP1 inhibitors can enhance their cytotoxic activity and reduce the conversion into single demethylated derivatives. Nobiletin and tangeretin can arrest MDA-MB-468 cells in the G1 phase, but the cell cycle of MCF10A cells is not significantly affected under the same conditions (*Androutsopoulos et al., 2009*; *Surichan et al., 2018a*; *Surichan et al., 2018b*). These studies suggest that nobiletin, tangeretin, and sinensetin can inhibit the activity and expression of CYP1 family enzymes in breast cancer cells, promote their own metabolism, and enhance the anti-proliferation activity of themselves or other anticancer drugs.

### Arylhydrocarbon receptor inhibitor

The arylhydrocarbon receptor (AhR) has a dual role in carcinogenesis and tumor suppression. Studies by *Benoit et al. (2022)* show that AhR activation is linked to breast cancer cell death through adverse outcome pathways analysis. However, AhR activation can also promote tumor growth, angiogenesis, migration, and metastasis while reducing apoptosis, inflammation, endothelial cell migration, and invasion. The prognostic value of AhR for breast cancer patient survival is still debated, and AhR function varies across different breast cancer cell lines. Endogenous, partially synthesized, and halogenated aromatic hydrocarbon AhR ligands may inhibit breast cancer proliferation through AhR-dependent degradation of ER, while some ligands like phthalates can promote breast cancer (*Safe & Zhang, 2022*). Additionally, cabazitaxel degrades ERα *via* the AhR-dependent proteasome pathway in ER[+] breast cancer cells (*Chen et al., 2022*).

3′,4′-DMF is an AhR antagonist in breast cancer cells. It hinders the transformation of AhR induced by 2,3,7,8-tetrachlorodibenzo-p-dioxin and the formation of AhR nuclear complex (*Lee & Safe, 2000*). Furthermore, 3′,4′-DMF, 5,7-DMF, and 4′,7-DMF can work as potential AhR inhibitors to prevent and/or treat hormone-dependent cancers (*Ibrahim & Abul-Hajj, 1990*; *Van Lipzig et al., 2005*; *Ta & Walle, 2007*). Also, 6,2′,4′-TMF acts as an AhR antagonist that competes with agonists, diminishing AhR's affinity for homologous response elements and agonist-dependent AHR signaling, like the transactivation of endogenous targets (*e.g.*, CYP1A1) (*Murray et al., 2010*). *Weiss et al. (2020)* discovered nobiletin, tangeretin, and sinensetin decrease CYP1A1/CYP1A2 activity through AhR activation (*Weiss et al., 2020*), however, it is unclear whether these PMFs act as AhR ligands to activate or inhibit AhR and suppress breast cancer through downstream signaling. The effectiveness of PMFs as AhR agonists or antagonists varies widely and may depend on the methoxy substitution sites and/or numbers of flavonoid precursor structures and the heterogeneity of breast cancer.

### Interferes with the biosynthesis of steroid hormones

ER[+] breast cancer comprises 80% of all breast cancers (*Zardavas et al., 2015*; *Siersbæk, Kumar & Carroll, 2018*), and the estrogen signaling pathway and ER play critical roles in its development (*Hanker, Sudhan & Arteaga, 2020*). AhR is overexpressed in ER[+] breast tumors, and various CYP enzymes are involved in estrogen synthesis and metabolism (*Blackburn et al., 2015*). Plant estrogens, like dietary PMFs, can act as substrates for CYP1s family enzymes, potentially interfering with steroid hormone biosynthesis, reducing endogenous hormone levels, and depriving breast cancer of important growth factors (*Henderson & Feigelson, 2000*; *Arroo et al., 2014*). *Hermawan & Putri (2020)* used bioinformatics analysis to demonstrate that nobiletin targets estrogen signaling (*Hermawan & Putri, 2020*). Nobiletin suppresses the conversion of testosterone into 17β-estradiol *via* inhibiting AhR activity and expression in MCF-7 cells (*Rahideh et al., 2017*). Tangeretin significantly inhibits breast cancer growth by lowering hormone and corresponding receptor expression (*Lakshmi & Subramanian, 2014a*). Tangeretin has no estrogenic activity but competes with 17β-estradiol to bind to ER, thereby eliminating its estrogenic effects in the presence of tamoxifen, an ER antagonist (*Stroheker et al., 2004*). Although some PMFs lack relevance in the literature, studies suggest that PMFs may have anticancer effects by influencing steroid hormone synthesis.

### Chemical sensitizers

Drug absorption in the intestine and liver heavily relies on ABC transporters such as P-gp (P-glycoprotein), BCRP (breast cancer resistance protein), and MRP1 (multi-drug resistance-related protein1). These transporters increase the excretion of anticancer drugs in cancer cells and effectively reverse drug resistance. Endocrine therapy drug resistance in breast cancer treatment poses a significant challenge to clinicians and threatens clinical benefits. Sensitizers for combination therapy can overcome drug resistance and improve efficacy (*Prajapati, Gupta & Kumar, 2022*). The PMFs discussed in the article not only act as combination therapy sensitizers with anticancer activity but also have low toxicity, better metabolic stability, and a more ready availability of resources.

   5,7-DMF and 7,4′-DMF inhibit BCRP and increase the cytotoxicity of several drugs by inhibiting their excretion mediated by BCRP (*An, Wu & Morris, 2011*; *Bae et al., 2018*; *Fan et al., 2019*). 3′,4′,7-TMF also effectively reverses BCRP-mediated drug resistance but has low anti-P-gp and no anti-MRP1 activity (*Katayama et al., 2007*; *Tsunekawa et al., 2019*). 3,3′,4′,5,6,7,8-HMF has also been shown to act as a BCRP inhibitor (*Pick et al., 2011*). Tangeretin and nobiletin are potent BCRP inhibitors that significantly increase the accumulation of several drugs (Paclitaxel, Doxorubicin, Docetaxel, Dunobicin and Dasatinib) in cells (*Fleisher et al., 2015*; *Ma et al., 2015*). Structural characteristics such as hydrophobic groups and hydrogen bond acceptors on the aromatic B-ring, double bonds between positions 2 and 3, and the 3-methoxy group in PMFs positively contribute to BCRP inhibition. Molecular docking analysis suggests potential interactions through pi-pi stacking and/or pi-alkyl interactions (*Pick et al., 2011*; *Fan et al., 2019*).

   7,3′,4′-TMF, 3,3′,4′,5,6,7,8-HMF, sinensetin, nobiletin, and tangeretin inhibit drug efflux *via* P-gp and MRP2, increasing drug bioavailability and reversing drug resistance.

The list of drugs affected include paclitaxel, colchicine, methotrexate, silybin, talinolol, and rhodamine-123 (*Takanaga et al., 2000a*; *Choi, Kim & Kim, 2004*; *Honda et al., 2004*; *Jeong & Choi, 2007*; *Mertens-Talcott et al., 2007*; *Yuan et al., 2018*; *Feng et al., 2020*). Several studies have identified specific structural characteristics of PMFs that contribute to P-gp inhibition, including hydrophobicity in the A/C and B rings, 6-methoxy in the A ring, and 3′-methoxy and 4′-methoxy in the B ring, as well as the 5′-methoxy substitution. The 6-methoxy group may provide steric hindrance or cause repulsion with the P-gp pocket wall. The position and number of methoxy groups are more important than hydroxylation. Molecular docking analysis suggests pi-pi stacking or pi-alkyl interactions facilitate binding to P-gp (*Choi, Kim & Kim, 2004*; *Fang et al., 2019b*; *Bai et al., 2019*).

Nobiletin inhibits MRP1 efflux transporters (*Nabekura, Yamaki & Kitagawa, 2008*), while tangeretin inhibits MRP2 efflux transporters (*Yuan et al., 2018*). Studies have shown that PMFs with a 4′-methoxy substitution are more likely to bind to MRP2, while those with a 3′-methoxy substitution are less favorable (*Fang et al., 2019a*).

The PMFs can also act as chemosensitizers in breast cancer *via* other mechanisms. The combination of nobiletin and tamoxifen increases sensitivity of T47D cells to tamoxifen. Nobiletin enhances the toxicity of doxorubicin in MCF7 cells but has no effect on T47D cells. Tangeretin and docetaxel may induce cell death by p53-dependent apoptosis in both MCF7 and T47D cells (*Meiyanto et al., 2011*). Additionally, tangeretin enhances the anticancer activity of metformin in different breast cancer cell lines (MCF7, MDA-MB-231, and their drug-resistant phenotypes) by generating reactive oxygen species, inhibiting cell migration, inducing cell cycle arrest, and promoting cancer cell apoptosis (*Mdkhana et al., 2021*). However, adding tangeretin to the drinking water of MCF7/6 tumor-bearing mice neutralizes the beneficial tumor-suppressive effects of tamoxifen, possibly due to decreasing natural killer cell numbers (*Bracke et al., 1999*; *Bracke et al., 2002*; *Depypere et al., 2000*).

PMFs have advantages over other efflux transporter inhibitors, including low toxicity, better biometabolic stability, and tissue distribution. They can potentially reverse drug resistance and increase drug bioavailability. While some studies suggest that PMFs can enhance anticancer activity by inhibiting efflux transporters and inducing apoptosis and cell cycle arrest, concerns remain about their effectiveness when used alone. For example, tangeretin's growth inhibition of tamoxifen is neutralized *in vivo*, possibly due to its biometabolic or estrogen-like effects. These issues can be addressed through new targeted drug delivery systems and increased dosage. Despite these challenges, PMFs have potential as a novel and promising chemosensitizer in breast cancer drug combination therapy.

### Others

These PMFs (5,6,7,3′,4′,5′-hexamethoxyflavone, 3,5,6,7,8,3′,4′-HMF and sinensetin) induce apoptosis, regulate the cell cycle, and inhibit angiogenesis, thereby suppressing proliferation, migration, invasion, and sphere formation in breast cancer. In addition to nobiletin and tangeretin, 3,5,6,7,8,3′,4′-HMF activate Ca (2+)-dependent apoptotic proteases to inhibit the growth of human breast cancer cells (*Sergeev et al., 2006*). 5,6,7,3′,4′,5′-hexamethoxyflavone can arrest triple-negative breast cancer cells in the

G2/M phase by inhibiting phosphorylation levels of signaling molecules in the MAPK and Akt pathways, thereby regulating cell proliferation and the cell cycle (*Borah et al., 2017*).

Nobiletin has anti-tumor activity against different types of breast cancer. In triple-negative breast cancer, nobiletin inhibits ERK activity, induces cell cycle arrest in G1/G2 phase, induces apoptosis, suppresses AKT and downstream mTOR activity, and inhibits ROR nuclear receptors to exert its anti-tumor effects. Nobiletin can be used alone or in combination with chemotherapeutic drugs like docetaxel (*Morley, Ferguson & Koropatnick, 2007b*; *Chen et al., 2014b*; *Kim et al., 2022*). In estrogen receptor-positive breast cancer, nobiletin exerts anti-tumor effects by inhibiting NF-κB signaling (*Chen et al., 2014a*; *Liu et al., 2018*; *Kim et al., 2022*), suppressing MMP-2, MMP-9, and CXCR4 expression, inhibiting MMP-9 enzymatic activity to suppress metastasis (*Ahn et al., 2012*; *Liu et al., 2018*), inhibiting angiogenesis *via* Src-FAK, STAT3, CD36/Stat36/NF-κB signaling (*Nipin et al., 2017*; *Sp et al., 2018*), regulating miR-200b/JAZF1/NF-κB signaling axis and Bax, Bcl-2, p53 proteins to induce apoptosis and autophagy (*Liu et al., 2018*; *Wang et al., 2021a*), and inhibiting IL-7-induced ERK and JNK signaling to suppress invasion and migration *in vivo* (*e.g.*, xenograft tumors and liver metastases) and *in vitro* studies (*Wu et al., 2023*). Data mining and network pharmacology suggest that nobiletin has potential targets for breast cancer treatment, including ESR1, MYC, CCND1, EGFR, and ERBB2, and related signaling pathways such as PI3K-AKT, p53, and ERBB (*Yang et al., 2021*).

Tangeretin has also anti-tumor activity against various types of breast cancer. It inhibits the Stat3 signaling pathway, induces apoptosis, and inhibits cell proliferation and tumor growth in breast cancer stem cells (*Ko et al., 2020*). In triple-negative breast cancer, tangeretin causes G1 phase cell cycle arrest, inhibits CDK2/CDK4 activity, and induces CDK inhibitors like p27 and p21 (*Morley, Ferguson & Koropatnick, 2007b*). In MDA-MB-231 resistant tumor cells, tangeretin induces apoptosis, blocks cell cycle arrest in G2/M phase. In estrogen receptor-positive breast cancer, tangeretin inhibits ERK phosphorylation, induces p53-independent apoptosis (*Van Slambrouck et al., 2005*), enhances the E-cadherin/catenin complex function, and inhibits extracellular matrix adhesion and invasion. It also downregulates IL-2 receptors on T lymphocytes and natural killer cells (*Bracke et al., 2002*). Studies show that oral administration of tangeretin inhibits tumor growth and metastasis in experimental breast cancer rats. It regulates cell metabolic energy flux, reduces tumor cell proliferation markers such as PCNA, COX-2, and Ki-67, blocks the G1/S phase of tumor cells, and inhibits MMP-2, MMP-9, and vascular endothelial growth factor. At the same time, it improves oxidative stress in renal tissue and protects kidneys from oxidative damage (*Lakshmi & Subramanian, 2014b*; *Arivazhagan & Sorimuthu Pillai, 2014*; *Periyasamy et al., 2015*; *Periyasamy et al., 2016*). Tangeretin targets potential targets for breast cancer treatment, including MTOR, NOTCH1, TP53, MMP9, NFKB1, PIK3CA, PTGS2, and RELA. It may inhibit breast cancer cell metastasis by regulating the PI3K/AKT signaling pathway (*Hermawan et al., 2021*).

The anti-breast cancer mechanism of target PMFs is a complex interaction network (Fig. 2), and there are many mechanisms that have not been explored and reported, which

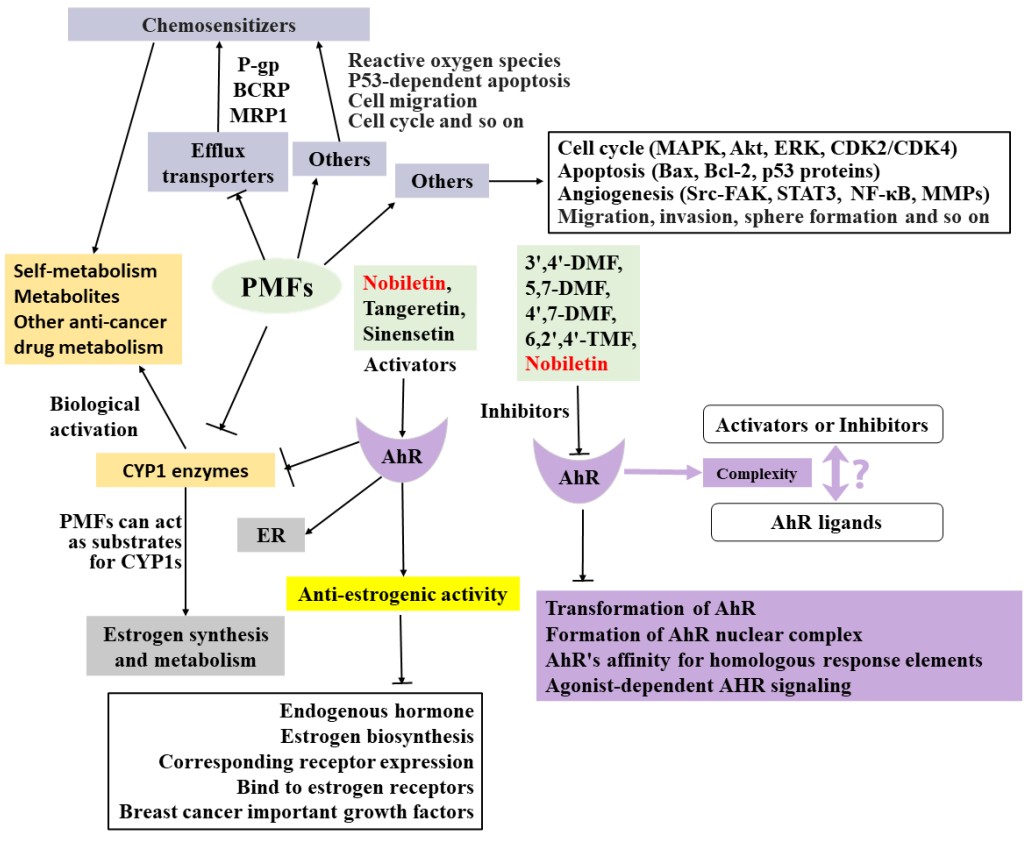

**Figure 2** Summary of the anti-breast cancer mechanism of targeted PMFs.

need to be further explored. Understanding the absorption and metabolism of PMFs in *vivo* is essential to clarify their anticancer effects.

## The details of administered PMFs
### The absorption and metabolism of orally administered PMFs in vivo

Most flavonoids, including PMFs, have excellent biological functions *in vitro*, but their limited bioavailability hinders reproduction *in vivo*. Intestinal absorption and liver metabolism are crucial factors affecting the systemic bioavailability and therapeutic efficacy of drugs after oral intake. Orally administered PMFs are likely absorbed in the small intestine, similarly to other drugs (*Ting et al., 2015*; *Wang et al., 2021b*).

Nobiletin is easily absorbed and metabolized due to its lipophilicity and high permeability (*Li et al., 2006*; *Singh et al., 2011*; *Zeng et al., 2017*; *Lan et al., 2021*). Nobiletin, tangeretin, and sinensetin commonly undergo Phase I metabolic reactions, including demethylation at the 4′position, and Phase II reactions involving conjugation with glucuronide (*Nielsen et al., 2000*; *Manthey et al., 2011*; *Zeng et al., 2017*; *Guo et al., 2021*; *Yu et al., 2022a*). Enzymes involved in demethylation and deglycosylation in intestinal cells may play a crucial role in determining bioavailability (*Németh et al., 2003*). However, there are variations in the types and amounts of metabolites produced from nobiletin and tangeretin, which
may be attributed to differences in individual gastrointestinal microbiota and/or drug metabolizing enzymes. Otherwise, no pharmacokinetic studies have been conducted on other target PMFs, highlighting the need for further research.

### Methylation contributes to the improvement of flavonoids bioavailability in vivo

Methylated flavonoids have better absorption rates, bioavailability, and metabolic stability than non-methylated flavonoids when they are orally administered, however, human data on this topic is lacking (*Tsuji, Winn & Walle, 2006*; *Ta & Walle, 2007*; *Walle, 2007b*; *Najmanová et al., 2020*). The PMF-metabolizing bacteria Blautiasp.MRG-PMF1 has demethylation and deglycosylation activities that can transform PMFs into various demethylated metabolites *via* PMF demethylase in the human intestinal tract (*Kim, Kim & Han, 2014*; *Burapan, Kim & Han, 2017*). 5,7-DMF and 7,4′-DMF are more effective at inhibiting cell proliferation and have higher oral bioavailability, resistance to metabolism in the liver metabolism, and tissue accumulation than their non-methylated analogs baicalein and apigenin (*Wen & Walle, 2006*; *Walle et al., 2007*). Long-term oral administration of nobiletin may have potential beneficial effects on regulating gut microbiota and improving PMF-demethylation ability (*Chen et al., 2020*; *Zhang et al., 2021*). The methoxy group in 3′,4′,5′,5,7-pentamethoxyflavone confers better oral metabolic stability and bioavailability (*Cai et al., 2011*). *Mullen, Edwards & Crozier (2006)* also found that orally administered dietary methylated flavonoids were absorbed faster in healthy human volunteers (*Mullen, Edwards & Crozier, 2006*).

PMFs have unique metabolic properties that differ from other flavonoids, leading to their wide distribution and accumulation in tissues (*Murakami et al., 2002*). Structural methylation may be a solution to optimize PMF's bioavailability while maintaining their biological activity as dietary flavonoids. Methylation of flavonoids can be optimized by introducing methyl groups at key sites through external chemical modification or enhancing metabolism through absorption and metabolism optimization. However, further research is needed to determine whether increasing the number of methyl substitutions can enhance the benefits of methylated flavonoids, and to find the optimal number and substitution sites for effective methylation.

### Metabolic activation or inactivation of administered PMFs by P-450 enzyme system

Xenobiotic substances, including PMFs, are primarily metabolized in the liver through the cytochrome P-450 system, with P-450 enzymes playing a key role in activating PMF metabolism. Some studies suggest that PMFs have greater resistance to P-450 dependent metabolism compared to their hydroxylated derivatives. Inhibitory mechanisms of various flavonoid derivatives on CYP1A1, 1A2, 1B1, 2C9, and 3A4 differ, with the number and position of hydroxyl and/or methoxy groups influencing their effect on these enzymes (*Walle, 2004*; *Shimada, 2017*). Hydroxyl and/or methoxy substitutions at the 3′ and/or 4′ positions of flavonoids are factors that affect the selectivity of different cytochrome P-450 enzymes (*Doostdar, Burke & Mayer, 2000*; *Walle, 2004*; *Shimada, 2017*). The inhibitory effects of PMFs on CYP enzymes may depend on the treatment concentration, duration, and

the number of methoxy substitutions on the A-ring of the flavonoid structure. However, these effects require further investigation and validation.

### Species source, extraction, identification, and toxicity of administered PMFs

The relevant plant species sources (http://bidd.group/NPASS/) of all target PMFs that can be retrieved are presented mainly searched by NPASS and other literature database (*Zhao et al., 2023*). Target PMFs can be derived from a variety of species, not limited to Rutaceae (Table 1). Target PMFs were mostly extracted using methanol or ethanol or water or mixtures thereof, and individual studies have obtained target products by other methods such as pressurized water, petroleum, organic solvents extraction or supercritical fluid chromatography. Their separations were mainly through RP-HPLC and ELSD, and identifications were mainly by IR; NMR (1H and 13C); MS; GC-MS and GC-MS, and a few included others such as melting points. Toxicity information collected by NPASS database in Table 2. From the data, it is found that the overall trend of target PMFs toxicity is a certain decrease with the increase of the number of methoxy (*Green et al., 2013*; *Mawatari et al., 2023*). The target PMFs all have certain drug-induced liver damage, but many studies have shown that PMFs can effectively improve liver physiology, especially those derived from citrus. This may be due to a lack of toxicity algorithm models. By modification of substituents or in combination with other compounds, PMFs are increased in bioavailability, reducing toxicity, further enhancing their biological activity, including anti-cancer ability. This may be a promising research direction for fully utilizing species resources containing target PMFs in relevant corresponding regions.

## DISCUSSIONS AND FUTURE

PMFs exhibit anti-steroidal hormone activity in $ER^+$ breast cancer, which may make them useful as endocrine therapy for inhibiting other hormone-dependent cancers such as prostate, ovarian, and uterine cancers, particularly those that are estrogen-dependent. Hormone therapy is known to significantly improve patient survival rates for breast and prostate cancers, which are the most common hormone-dependent cancers in women and men, respectively. Flavonoids sourced from the diet, known as plant estrogens, can exert various activities, including estrogenic, anti-estrogenic, non-estrogenic, and biphasic, through different mechanisms such as binding ligands to ER, regulation of selective estrogen receptor modulator activity, endocrine disruption, and interference with steroid biosynthesis/metabolism and other cell signaling pathways (*Kiyama, 2023*). Additionally, flavonoids can interfere with testosterone synthesis by inhibiting HSD3B activity (*Zhang et al., 2019*). Chrysin, a type of flavonoid, was found to selectively induce proteasomal degradation of AR-V7 but not AR by *Liu et al. (2021)*. Hydroxyl and isopentenyl groups were found to be essential for flavonoid estrogenic activity based on QSAR analysis, while methylation of hydroxyl and cyclization of isopentenyl significantly reduce estrogenic activity (*Zhang et al., 2018*). Coumestrol, due to its two hydroxyl groups' positions aligning with estradiol (*Torrens-Mas & Roca, 2020*), suggests that PMFs undergo *in vivo* metabolism, which may enhance or attenuate their estrogenic activity. In this study, PMFs were found to have anti-breast cancer activity mainly focused on anti-estrogenic activity, with little

**Table 1  Details of PMFs with anti-breast cancer activity.**

| Names | Species and plant parts |
|---|---|
| 3′,4′-DMF | *Agelaea pentagyna* leaves (*Kuwabara et al., 2003*); *G. biloba* leaves (*Rajaraman et al., 2009*); *Primula veris*. |
| 5,7-DMF | *Leptospermum scoparium* (*Häberlein, Tschiersch & Schäfer, 1994*); *Piper methysticum* roots (*Wu, Nair & De Witt, 2002*); *Kaempferia parviflora* rhizomes (*Tep-Areenan, Sawasdee & Randall, 2010*); *Tripterygium wilfordii* and so on. |
| 7,4′-DMF | Not found. |
| 7,3′,4′-TMF | *Myroxylon peruiferum* (*Ohsaki et al., 1999*). |
| 6,2′,4′-TMF | Leaves and stems of *T. acutifolius* (*Apaza et al., 2019*); aerial parts of *Artemisia campestris subsp. glutinosa (Besser) Batt.* (*Apaza Ticona et al., 2020*). |
| Tangeretin | Orange juice (*Takanaga et al., 2000a*; *Takanaga et al., 2000b*); pericarpium of *Citrus reticulata* (cv Jiao Gan) (*Mak et al., 1996*); Citrus peel (*Li et al., 2007*; *Wang et al., 2007*); *Citrus grandis* Osbeck leaves (*Kim et al., 2010*) and so on. |
| Sinensetin | Green tangerine peel of *Pericarpium Citri Reticulatae Viride* (*Wang et al., 2007*); *Citrus grandis* Osbeck leaves (*Kim et al., 2010*), etc. |
| Nobiletin | Orange juice (*Takanaga et al., 2000a*; *Takanaga et al., 2000b*); pericarpium of *Citrus reticulata* (cv Jiao Gan) (*Mak et al., 1996*); *Citrus sinensis* peel (*Li et al., 2007*); *Citrus grandis* Osbeck leaves (*Kim et al., 2010*), etc. |
| 5,6,7,3′,4′,5′- HexMF | *Ageratum conyzoides L.* leaves (*Faqueti et al., 2016*); aerial parts of *Eremophila debilis* (Myoporaceae) (*Butler et al., 2018*), etc. |
| 3,3′,4′,5,6,7,8-HMF | Orange juice (*Takanaga et al., 2000a*; *Takanaga et al., 2000b*); Citrus peel (*Li et al., 2007*; *Wang et al., 2007*); *Murraya paniculate* (*Sangkaew et al., 2020*). |

**Notes.**
DMF, dimethoxyflavone; TMF, trimethoxyflavone; PMF, pentamethoxyflavone; HexMF, hexamethoxyflavone; HMF, heptamethoxyflavone.

**Table 2  Toxicity of PMFs with anti-breast cancer activity.**

| Names | H-HT | DILI | ROAT | MRDD | SkSe | Carc | EyCo | EyIr | ReTo |
|---|---|---|---|---|---|---|---|---|---|
| 3′,4′- DMF | 0.128 | 0.921 | 0.16 | 0.151 | 0.434 | 0.73 | 0.004 | 0.699 | 0.392 |
| 5,7-DMF | 0.129 | 0.855 | 0.069 | 0.52 | 0.621 | 0.249 | 0.009 | 0.915 | 0.59 |
| 7,3′,4′-TMF | 0.166 | 0.938 | 0.098 | 0.566 | 0.548 | 0.284 | 0.004 | 0.439 | 0.406 |
| Tangeretin | 0.096 | 0.949 | 0.352 | 0.024 | 0.257 | 0.083 | 0.003 | 0.08 | 0.032 |
| Sinensetin | 0.076 | 0.85 | 0.159 | 0.049 | 0.383 | 0.035 | 0.003 | 0.04 | 0.198 |
| Nobiletin | 0.07 | 0.915 | 0.326 | 0.02 | 0.23 | 0.052 | 0.003 | 0.043 | 0.031 |
| 5,6,7,3′,4′,5′- HexMF | 0.056 | 0.705 | 0.165 | 0.045 | 0.466 | 0.027 | 0.003 | 0.055 | 0.128 |

**Notes.**
H-HT, human hepatotoxicity; DILI, drug-inuced liver injury; ROAT, rat oral acute toxicity; MRDD, maximum recommended daily dose; SkSe, skin sensitization; Carc, carcinogencity; EyCo, eye corrosion; EyIr, eye irritation; ReTo, respiratory toxicity.

emphasis on anti-androgenic activity. Combining PMFs' estrogenic activity with inhibitory resistance proteins activity indicates their potential to against resistance to endocrine and chemotherapy for hormone-dependent tumors. More research and clinical validation are necessary to realize this potential.

PMFs have been found to inhibit the growth of breast cancer directly or indirectly through various mechanisms, such as inhibiting resistance proteins, inducing apoptosis, and regulating cell cycle progression. Current research has focused on how to improve the bioavailability of PMFs, with researchers exploring methods such as methylation conversion, new formulations, and delivery systems to enhance anti-cancer ability. Methylated flavonoids show better anticancer ability, and *Lee et al. (2017)* used genetic engineering microorganisms to produce DMF compounds effectively from non-methylated flavonoid precursors (*Lee et al., 2017*). Various drug delivery systems, such as stable picolinic emulsions with corn protein/prulanan polysaccharide composite colloidal particles, amorphous solid dispersions, nanoemulsions, water-resistant polyvinyl alcohol/polyacrylic acid electrospun fibers, plant exine capsules, and hydrophobic nanoparticles have been utilized to improve the release, dissolution, liver delivery, concentration–time curve area, and the bioavailability of PMFs (*Onoue et al., 2013*; *Hattori et al., 2019*; *Wu et al., 2020*; *Wang et al., 2022*; *Qu et al., 2022*; *Yu et al., 2022b*). Additionally, RGD peptide-modified nano-carriers and tangeretin-zinc oxide quantum dots have targeted cancer cells by taking advantage of pH sensitivity and the acidic pH maintained in the tumor microenvironment, significantly inhibiting tumor growth *in vivo* (*Roshini et al., 2018*; *Bao et al., 2020*; *Zhan et al., 2021*). Further research is needed to determine the anti-breast cancer activity of other PMFs and their metabolites, as well as their effectiveness against different types of cancer and relevant molecular mechanisms.

PMFs have been shown to have some efficacy in the treatment of breast cancer. At the cellular level, they can enhance the sensitivity of chemotherapy and endocrine therapy through combination therapy. However, at the systemic level, combination therapy with PMFs can hinder endocrine therapy. Further studies should be conducted in order to improve the clinical potential of PMFs for breast cancer treatment. Additional statistical comparisons of the *in vivo* anti-tumor activity of PMFs, investigation of any corresponding relationship between the heterogeneity of breast cancer and the diversity of PMF types, and improved clinical trial data are needed. Research on PMFs' anti-breast cancer properties has increased in recent years, suggesting unknown anti-breast cancer mechanisms of PMFs. Clinical trials and sample collections have been limited so far, with only one relevant clinical trial (NCT00702858) retrieved from ClinicalTrials.gov and the WHO International Clinical Trials Registry Platform. This trial sought to determine the effect of blue citrus fruit to reduce joint and bone pain associated with aromatase inhibitor treatment in postmenopausal $ER^+$ breast cancer patients and its effect on the quality of life for these patients (*Johnson, 2023*).

## CONCLUSIONS

Although the anti-breast mechanism of the target PMFs has been thoroughly explored to form an interactive network (Fig. 2), further studies are needed. As anticancer agents, PMFs

have various advantages, such as abundant resources and easy availability, homologous medicinal and food sources, and low toxicity and side effects. Overall, PMFs have significant potential for preventing or treating breast cancer and have promising clinical applications.

## ACKNOWLEDGEMENTS

We acknowledge the editors and anonymous reviewers for insightful suggestions on this work.

### Funding

The authors received no funding for this research. The Xinjiang Youth Science and Technology Top-notch Talent Project-Youth Science and Technology Innovation Talent Training (2022TSYCCX0022)) paid for the APC. The funders had no role in study design, data collection and analysis, decision to publish, or preparation of the manuscript.

### Grant Disclosures

The following grant information was disclosed by the authors:
The Xinjiang Youth Science and Technology Top-notch Talent Project-Youth Science and Technology Innovation Talent Training: 2022TSYCCX0022.

### Competing Interests

The authors declare there are no competing interests.

### Author Contributions

- Yiyu Wang conceived and designed the experiments, performed the experiments, analyzed the data, authored or reviewed drafts of the article, and approved the final draft.
- Yuan Mou analyzed the data, prepared figures and/or tables, and approved the final draft.
- Senlin Lu analyzed the data, prepared figures and/or tables, and approved the final draft.
- Yuhua Xia analyzed the data, prepared figures and/or tables, and approved the final draft.
- Bo Cheng conceived and designed the experiments, authored or reviewed drafts of the article, and approved the final draft.

### Data Availability

  This is a literature review.

### Supplemental Information

Supplemental information for this article can be found online at http://dx.doi.org/10.7717/peerj.16711#supplemental-information.

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
