# Peer review of "Polymethoxylated flavonoids in citrus fruits: absorption, metabolism, and anticancer mechanisms against breast cancer"

_PeerJ, doi:10.7717/peerj.16711_

## Round 0.1 · original submission · Major Revisions

Dear authors,

Please read the comments and recommendations of our reviewers carefully. Please indicate all changes in the revision. The reviewers will check your revision; thus make sure that you follow their advice.

Kind regards
Michael Wink
AE

**Language Note:** The review process has identified that the English language must be improved. PeerJ can provide language editing services - please contact us at copyediting@peerj.com for pricing (be sure to provide your manuscript number and title). Alternatively, you should make your own arrangements to improve the language quality and provide details in your response letter. – PeerJ Staff

Reviewer 1 ·

Basic reporting

This is an interesting mini-review that discusses the absorption, metabolism, and anti-breast cancer properties of 10 selected polymethoxylated flavones derived from citrus fruits.

Experimental design

The authors reviewed the available literature using several scientific research engines and discussed them.

Validity of the findings

• In the search strategy section, the authors could enhance the clarity of their approach by offering more comprehensive details. For instance, it would be beneficial if you specify the total number of hits found and how these were further assessed for relevance to your review. The authors may also consider providing the keywords they used during the search. Next, they may describe what criteria were used to include or exclude publications from the review, e.g., not relevant to the aim of your review, poor quality, etc. The authors may consider using the Flow chart to improve the search strategy section. Likewise, it is also advisable to include references to existing reviews addressing the same topic or closely related subjects in the introduction, outlining the contributions of the current review compared to previously published works.
• While the authors have discussed the anti-breast cancer activity of 10 polymethoxylated flavonoids, they haven't included information about the corresponding citrus species of origin, the relevant plant parts, or the extraction and identification techniques employed. To enhance the manuscript's comprehensiveness, we recommend adding a section that briefly discusses the specific citrus species, extraction and identification methods, and plant parts utilized in obtaining the 10 discussed Polymethoxylated Flavonoids." A figure highlighting those structures will be also good.
• The manuscript is poor in terms of the toxicity of these compounds. The authors may consider adding a brief section, in which they could briefly discuss the toxicity/safety of these polymethoxylated flavonoids.
• The authors discussed the role of P450 as a potential activator of breast cancer through activating pro-and carcinogenic substances, emphasizing the role of these flavones in inhibiting P450 (as a therapeutic approach to prevent breast cancer), and also through the inhibition of AhR, etc. Could you please clarify whether the targeted compounds in your review affect intrinsic apoptosis pathways (e.g., cytochrome C, Bax/Bcl-2 ratio, caspase activation) and extrinsic pathways (P53, Death Receptors), anti-angiogenic (VEGF), within breast cancer cell lines?

Additional comments

The manuscript needs English editing

Reviewer 2 ·

Basic reporting

1. English correction is required.

2. Review the manuscript carefully to avoid errors,
For eg: Line 36 “Add your introduction here”

3. If possible, remove the “Intended Audience” section.

Experimental design

1. Since the major focus of the review is on the anti-cancer action of PMF, it is better to “ change “Anti-breast cancer molecular targets and mechanism of PMFs” as the first section.

2. There is no correlation between Pharmacokinetics section and Anti-cancer section. Modify the pharmacokinetics sections related to breast cancer research.

3. Include a diagram to represent the effect of PMF on different breast cancer signaling pathways.

Validity of the findings

Conclusion is missing

---

## Round 0.2 · accepted · Accept

Dear authors

Thank you for the revision of your manuscript, which addresses most concerns of the reviewers.

Thus I have good news for you, that we can accept the manuscript now.

Kind regards
M. Wink
AE

Reviewer 1 ·

Basic reporting

This is an interesting mini-review that discusses the absorption, metabolism, toxicity, and anti-breast cancer properties of 10 selected polymethoxylated flavones derived from citrus fruits.

Experimental design

The authors reviewed the available literature using several scientific research engines and discussed them.

Validity of the findings

no comment

Additional comments

no comment